

# FFENet: frequency-spatial feature enhancement network for clothing classification

Feng Yu[1,2], Huiyin Li[1], Yankang Shi[1], Guangyu Tang[1], Zhaoxiang Chen[1] and Minghua Jiang[1,2]

[1] School of Computer Science and Artificial Intelligence, Wuhan Textile University, Wuhan, Jiangxia District, China
[2] Engineering Research Center of Hubei Province for Clothing Information, Wuhan, Jiangxia District, China

## ABSTRACT

Clothing analysis has garnered significant attention, and within this field, clothing classification plays a vital role as one of the fundamental technologies. Due to the inherent complexity of clothing scenes in real-world environments, the learning of clothing features in such complex scenes often encounters interference. Because clothing classification relies on the contour and texture information of clothing, clothing classification in real scenes may lead to poor classification results. Therefore, this paper proposes a clothing classification network based on frequency-spatial domain conversion. The proposed network combines frequency domain information with spatial information and does not compress channels. It aims to enhance the extraction of clothing features and improve the accuracy of clothing classification. In our work, (1) we combine the frequency domain information and spatial information to establish a clothing feature extraction clothing classification network without compressed feature map channels, (2) we use the frequency domain feature enhancement module to realize the preliminary extraction of clothing features, and (3) we introduce a clothing dataset in complex scenes (Clothing-8). Our network achieves a top-1 model accuracy of 93.4% on the Clothing-8 dataset and 94.62% on the Fashion-MNIST dataset. Additionally, it also achieves the best results in terms of top-3 and top-5 metrics on the DeepFashion dataset.

## INTRODUCTION

With the rapid popularization of online shopping in the clothing industry, efficient clothing image classification (*Shajini & Ramanan, 2022*) can not only realize the automatic classification of clothing, but also greatly improve the efficiency of clothing retrieval and virtual try-on. The complexity and variety of clothing scenes pose a challenge for accurate clothing classification, which is an urgent issue to address in the application of clothing images in real-world scenarios.

Corresponding author
Minghua Jiang, minghua-jiang@wtu.edu.cn

In the field of clothing classification, a significant amount of research has been conducted in the past. Clothing classification differs from other classification tasks due to certain similarities among different clothing categories and variations within the same clothing category, such as patterns and colors. Previous research on clothing classification can be broadly categorized into two main types: 1) traditional machine learning methods (*Zhou, 2022*; *Ölçer, Ölçer & Sümer, 2023*), and 2) deep neural network methods (*Hassan et al., 2022*; *Sun et al., 2022*; *Al Shehri, 2022*). In the realm of clothing classification using traditional machine learning methods, researchers often improve basic classifiers. For instance, *Zhang et al. (2016)* incorporated Histogram of Oriented Gradient (HOG) (*Déniz et al., 2011*) into the Example Support Vector Machine (E-SVM) (*Noble, 2006*) classifier to enhance robustness to lighting conditions and improve the accuracy of E-SVM in clothing classification. Others proposed improvements to the fusion of Scale Invariant Feature Transform (SIFT) (*Cheung & Hamarneh, 2009*) and HOG for ethnic clothing classification. Some researchers utilized texture features and modified SIFT to obtain Speeded Up Robust Features (SURF) (*Bay et al., 2008*) for clothing classification. The above-mentioned methodological enhancements represent only a fraction of the innovative approaches in the field of clothing classification using traditional machine learning algorithms. While traditional machine learning algorithms offer faster adaptation to different scenarios, they generally exhibit lower accuracy compared to deep neural network algorithms.

Currently, most clothing classification tasks are based on deep neural network methods, which involve modifying mainstream classification models to suit clothing-specific scenes. Researchers propose improved convolutional neural networks (CNNs) (*Kiranyaz et al., 2021*; *Pan, Gupta & Raza, 2023*) for clothing classification by adjusting the structure of the original CNN model and increasing the size of the convolutional kernels in the modified structure. In the clothing domain, many researchers enhance the effectiveness of clothing classification by modifying the neural network's architecture and integrating other technologies. For example, *Bai et al. (2019)* introduced Bidirectional Convolutional Recurrent Neural Networks, which efficiently handle message-passing in syntactic topology and generate regularized landmark layouts. Based on this network, two attention mechanisms are Landmark-Aware Attention and Category-Driven Attention, which were designed to enhance clothing category classification. In another study, *Hidayati et al. (2017)* performed classification on clothing objects in the images. By analyzing the characteristics and clothing features present in the clothing images, the authors addressed the challenge of limited clothing image samples to some extent. Additionally, they introduced a comprehensive full-body clothing dataset containing 3250 images. *Zhang et al. (2020)* proposed the trained two-branch ImageNet backbone is used to enhance image shape and texture extraction, and the shape features are extracted by landmark detection. The effect improvement was verified in clothing category classification and clothing attribute classification respectively.

These previous challenges have demonstrated that the performance of deep neural networks in clothing classification can be enhanced by adjusting the network structure and incorporating clothing-specific feature enhancements. Furthermore, these studies have shown that improving the representation of clothing outlines can lead to increased

accuracy in clothing classification. Despite these advancements, several challenges still persist in the clothing classification domain: (1) poor accuracy of clothing classification in complex scenes: Current methods face difficulties in accurately classifying clothing in complex scenes, where multiple factors such as lighting conditions, occlusions, and background clutter can affect the classification results. (2) Insufficient improvement through spatial image feature enhancement: While enhancing spatial image features has shown promise, it alone may not be sufficient to significantly improve the accuracy of clothing classification. Further research is needed to explore other complementary approaches. (3) Similarity among different clothing categories and variation within the same category: Different clothing categories often share similar parts, making discrimination challenging. Additionally, clothing within the same category may exhibit significant texture variations, further complicating the classification task. Addressing these challenges is crucial for advancing the accuracy and robustness of clothing classification methods in real-world scenarios.

We propose a clothing classification network based on a frequency-spatial feature enhancement framework to address the aforementioned challenges. The main idea of the framework is as follows: the image input to the network is converted from the spatial domain to the frequency domain using the discrete cosine transform (DCT) (*Pang et al., 2019*), then the information in different frequency domains is extracted, different frequency domains store different information. The image information is divided into high frequency information and low frequency information, where the high frequency information stores the contour information and detail information, and the low frequency information stores the texture information. Finally, the spatial information and frequency information are used to enhance the objective feature for improving the classification accuracy. Our main contributions are threefold:

- A novel clothing classification network is proposed to improve the accuracy with frequency information and optimal backbone network, that is, frequecy-spatial feature enhancement network for clothing classfication (FFENet). Our proposed optimal backbone network consists of effective convolutional modules and efficient channel attention (ECA) (*Wang et al., 2020*) modules. A large number of experiments indicate that our proposed method can achieve the best performance among state-of-the-art methods.
- The frequency domain enhancement module is proposed to extract high and low frequency information from the feature maps and transform this information from the frequency domain into a spatial domain image. This transformation does not lose the original information, but increases the number of feature maps, allowing the network to focus on both contour and texture information.
- By collecting some public complex scene clothing images on kaggle websites and shopping websites, combining with a small part of clothing data in the Deepfashion dataset (*Liu et al., 2016*), and manually filtering the collected images, we obtain a dataset of 8 classified clothing styles with 5,156 high quality images.

## RELATED WORK

The related work consists of two main parts: 1) application of frequency domain in the field of image classification, and 2) mainstream deep neural networks for classification.

### Application of frequency domain in the field of image classification

Spatial domain images can be classified directly by using trained neural networks, and good classification results can be obtained, but this approach does not fully exploit the information in the image, which is the frequency domain information implies in the image. There are many ways to extract frequency domain information from an image, such as the Fourier transform, discrete cosine transform, wavelet transform, and other methods. Frequency domain information, as an alternative representation of the spatial domain image, may contain information that is not used by the neural network and is useful for classification. Researchers have also conducted research into the use of frequency domain information extracted from images to complement image processing tasks when using deep learning techniques.

First, for DCT, *Qin et al. (2021)* studied the effect of partially compressed input images using the DCT algorithm on the performance of neural networks. They found that while the DCT algorithm reduces data redundancy, there is a risk of losing valuable features for network learning. In a similar vein, *Xu et al. (2020)* investigated the use of DCT transformation on original images followed by CNN classification. They demonstrated through experiments that DCT features obtained directly from the JPG format can be processed as effectively as the original image data using the same CNN architecture. The neural network architecture with DCT features performed on par with the one using original image data. *Borhanuddin et al. (2019)* approached the channel attention mechanism from a different perspective by considering frequency analysis. They showed that regular global average pooling can be seen as a special case of frequency domain feature decomposition and proposed a novel multi-spectral channel attention structure. *Liu et al. (2018)* proposed methods to compress and accelerate neural network training by focusing on weights and their connections. They explored a data-driven approach to remove redundancy in both spatial and frequency domains, enabling the network to discard more unnecessary weights while maintaining similar accuracy. They achieved this by obtaining an optimal sparse CNN in the frequency domain and reducing the computational burden of convolution operations through linear combinations of DCT basis convolutional responses. On a different note, *Gueguen et al. (2018)* presented a simple idea where they directly used JPG image processing to generate DCT coefficients. They modified the ResNet 50 (*He et al., 2016*) network to accommodate DCT coefficients as direct inputs and evaluated the performance of this model on the ImageNet dataset.

In addition to DCT, as a powerful time domain evaluation analysis method, wavelet transform can also provide additional frequency domain information for deep learning techniques. *Li et al. (2020)* used a nonlinear model and average pooling for wavelet transformation and proposed the wavelet scattering network. The first network layer of this network outputted SIFT-type descriptors, while the next layer provided complementary translation invariant information to improve classification. The network computed a

translation invariant image representation that was stable to deformation and preserved high-frequency information for classification. However, the network couldn't be easily transferred to other tasks due to strict mathematical assumptions. To address the problem of CNN being susceptible to noise interference (*i.e.*, small image noise causing drastic changes in the output), *Bruna & Mallat (2013)* used discrete wavelet transform to replace max-pooling, step convolution, and average pooling in order to enhance CNN. They proposed a universal discrete wavelet transform and its inverse transform layer suitable for all kinds of wavelets. These layers were utilized to design a wavelet ensemble CNN for image classification.

In addition, a number of variants based on the wavelet transform (*e.g.*, the contour wavelet transform) had also been studied accordingly. *Liu et al. (2020)* proposed a new network architecture called the contourlet convolutional neural network, which was designed to learn sparse and effective feature representations of images. The contour wave transform was first applied to obtain spectral features from the image. Then, the spatial spectral feature fusion method was used to integrate the spectral features into the CNN architecture, followed by statistical feature fusion to integrate the statistical features into the network. Finally, the fused features were classified to obtain the results.

## Mainstream deep neural networks for classification

The image classification task is the task of determining which categories in the category space the input image belongs to. There are two types of mainstream classification networks, one based on convolutional neural networks and the other based on Transformer. Each of these two types has its own advantages and disadvantages. Convolutional neural network-based models work better for both small sample datasets and large datasets, and the network inference is faster. The Transformer-based classification model may not work as well on small datasets as the convolutional neural network-based one, which consumes a lot of memory space. However, it performs well on very large datasets, and the network is less prone to overfitting.

Convolutional neural network based classification models include GoogleNet (*Szegedy et al., 2015*), ResNet, DenseNet (*Huang et al., 2017*), EfficientNet (*Tan & Le, 2019*), ConvNext (*Liu et al., 2022*), and EfficientNetV2 (*Tan & Le, 2021*), among which the EfficientNetV2 model has best accuracy and computing speed compared with many classification networks. The focus of this paper is to improve the performance of the network on a small sample dataset, so in this paper, CNN is used to build our clothing classification network. The transformer (*Dong et al., 2022*; *Hua et al., 2022*) based classification models, such as ViT and SwinT (*Liu et al., 2021*), are initially used in the field of natural processing, where the transformer framework based on the attention mechanism achieved good results. Later, *Vaswani et al. (2017)* introduces the transformer to the field of computer vision, which worked well in mega databases, so the improvement of the Transformer based classification models hung a boom.

## OUR METHOD

The main task of the approach in this paper is to construct a model for clothing classification on a small sample clothing dataset for complex scenes. The category of clothing depends greatly on the silhouette features and textural characteristics of the clothing. We have summarised these rules, and if we can extract and learn these corresponding features through some techniques, it will be of great help in clothing classification.

Based on the above discussion we propose our approach (FFENet), where from the perspective of frequency domain, texture information and contour information in spatial domain are the information of different frequency bands. So we convert the spatial domain images into frequency domain images, transform them into different spatial feature maps by selecting information from different frequency bands, and put the spatial feature map information into the network we build for learning to improve the accuracy of clothing classification.

### Network overview

Our proposed network structure is shown in Fig. 1. When the clothing images are inputted into the DCT frequency domain enhancement (DCT-FDE) module, they are first converted into the YCbCr format. Subsequently, the converted feature map is divided into blocks, and the information of each block is then transformed from the spatial domain to the frequency domain using DCT. The list of spatial domain feature maps is generated based on the frequency domain information at the corresponding position of each block. Finally, the generated feature maps are stitched together to obtain the feature map F. In order to delve deeper into the learned feature map information, we propose the clothing feature extraction (CFE) module to further explore the information within the feature map F.

In the CFE module, we first perform a $3 \times 3$ convolution operation, followed by the modified fused MBConv block and the modified MBConv block. Subsequently, the feature map output from the CFE module is fed into the classification header for classification. The classification header involves applying a $1 \times 1$ convolutional layer for channel alignment, followed by global average pooling. Finally, two fully connected layers are utilized. The first fully connected layer is responsible for obtaining the preliminary sequence, while the second fully connected layer is used to generate the final prediction result.

### DCT frequency domain enhancement module

The transformation process of the DCT-FDE module is depicted in Fig. 2, which first converts the input RGB image into ycbcr format to obtain the feature map $x_{ycbcr} \in R^{H \times W \times 3}$. Subsequently, $x_{ycbcr}$ is partitioned into a set of $4 \times 4$ patches to obtain $\{p_{i,j} \in R^{4 \times 4 \times 3} | 1 \le i \le H//4, 1 \le j \le W//4\}$, where the patches are three channels. A dense DCT transformation is performed on the image window for each one, and each patch is processed in the frequency domain to obtain, where represents the patch corresponding to a particular colour channel in $\{d_{i,j} \in R^{4 \times 4 \times 3} | 1 \le i \le H//4, 1 \le j \le W//4\}$. Here each value in the patch corresponds to the intensity of a particular frequency band. In order to extract the information of different frequency bands separately, we filter the frequency bands for the number of times of chunk size squared, taking the information of only one

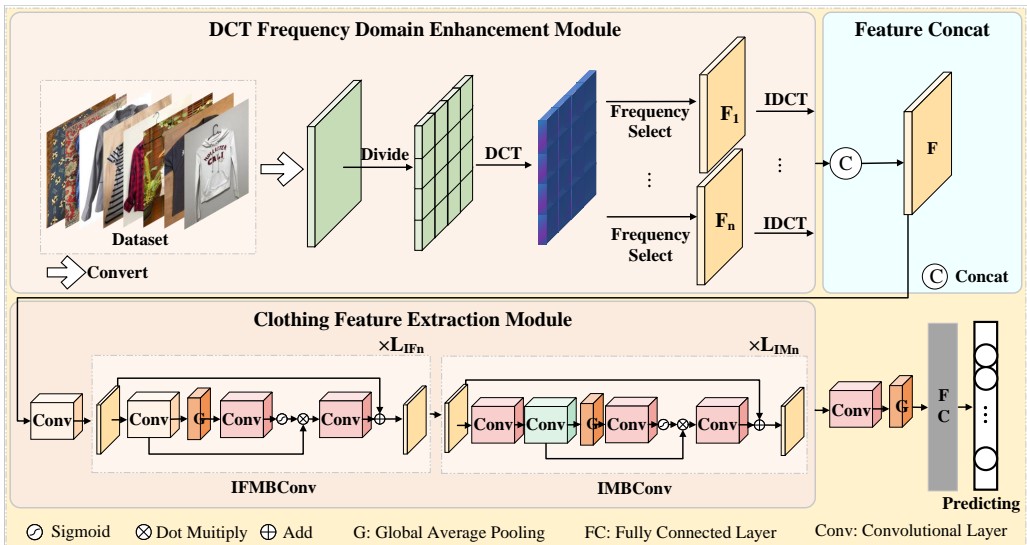

**Figure 1** **An overview of the proposed network.** We use the DCT frequency domain enhancement module to extract the spatial information of different frequency bands of the image, and use the stitching operation to concatenate all the spatial information feature maps to obtain F. The information of the feature map F is further extracted using the clothing feature extraction module, where $L_{IFn}$ and $L_{IMn}$ represent the number of repetitions of the corresponding layer of the IFMBConv block and IMBConv block, respectively. Finally, one $1 \times 1$ convolution operation, one global average pooling operation and two full connection operations are carried out in turn to obtain the final clothing classification results.

frequency band and filtering out the information of other frequency bands each time, and perform a DCT inverse transform to convert the filtered time-frequency domain information into spatial domain information after each filtering operation, and finally get a list of feature maps $x_1^{idct} \in R^{H \times W \times 3}, x_2^{idct} \in R^{H \times W \times 3}, \ldots, x_n^{idct} \in R^{H \times W \times 3}$, where the value of n is the square of the block size. We have a block size of 4 here, so we end up with 16 feature maps. As Fig. 3 shows an experiment we did, visualising the 16 feature maps obtained after inputting the image to the DCT-FDE module, we can see that the first band of the chunk stores the most colour and texture information, and that the other bands store more shape and detail information, which is what we call low-frequency information and high-frequency information. By this method we do not lose any information in any of the frequency bands, but it allows our subsequent proposed classification network to learn both high-frequency and low-frequency information, which in fact replaces the convolution operation in a sense and has a feature enhancement effect.

## Clothing feature extraction module

The MBConv block is a module unit originally introduced by MobileNetV2 (*Sandler et al., 2018*). It has since been widely adopted due to its portability and good performance. Subsequently, EfficientNet further built upon the MBConv block and achieved notable results in various tasks. EfficientNetV2, a follow-up work, suggests that employing fused MBConv blocks at the shallower layers of the network yields better performance. This improvement was achieved through the application of neural architecture search

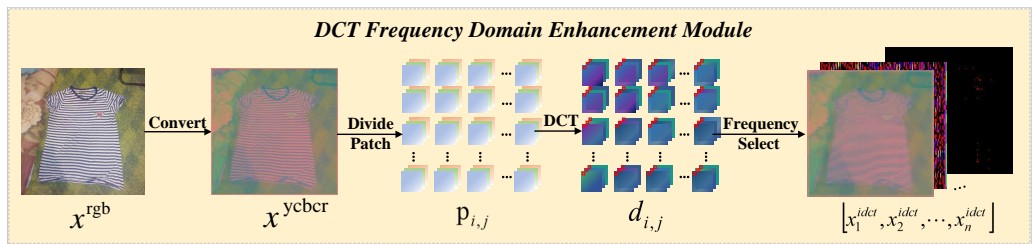

**Figure 2** The pipeline flow of the DCT frequency domain enhencement module.

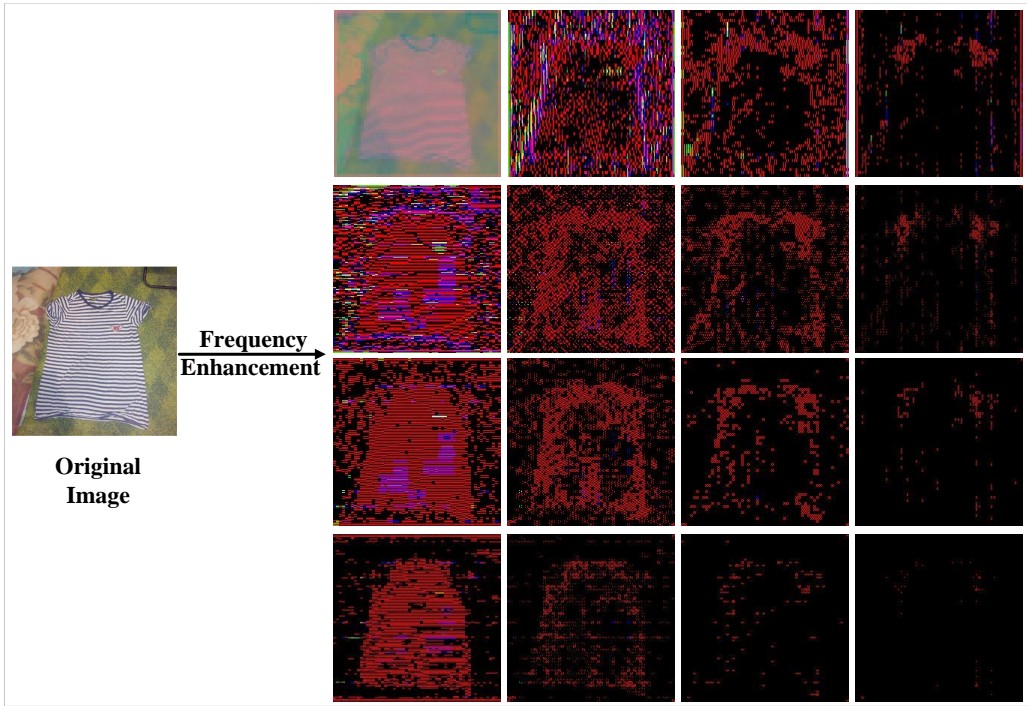

**Figure 3** The image after 4 × 4 DCT transformation and IDCT transformation of a single frequency band.

techniques, which allowed for the identification of more effective network configurations. The module we built is initially a combination of the fused MBConv block and the MBConv block, but in combination with the previous DCT-FDE module, we guess that there is channel compression in the squeeze-and-excitation (SE) module, which might lead to inadequate learning of our frequency domain information, so our DCT-FDE module improves the structure of the fused MBConv block and the MBConv block by replacing the SE module with the ECA module. Our experimental results prove our conjecture, please see the experimental section for details. We use ECA module to enforce the feature map, which is defined as follows:

$$F_t^{eca} = F_{t-1}(\sigma(Conv^{1\times1}(GAP(F_{t-1})))) \tag{1}$$

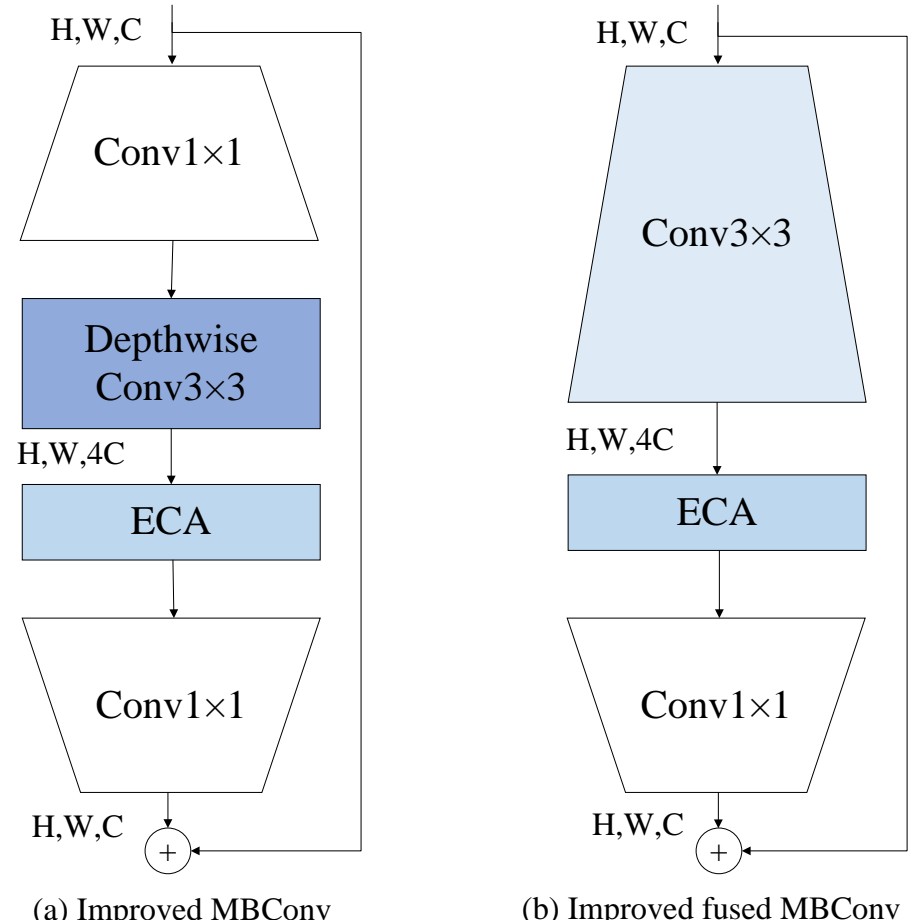

| H,W,C | | H,W,C |
| Conv1×1 | | Conv3×3 |
| Depthwise Conv3×3 | | |
| H,W,4C | | H,W,4C |
| ECA | | ECA |
| Conv1×1 | | Conv1×1 |
| H,W,C + | | H,W,C + |
| (a) Improved MBConv | | (b) Improved fused MBConv |

**Figure 4**  **The structure of the improved MBConv and the improved fused MBConv.**

where $F_{t-1}$ and $F_t^{eca}$ are the input and output feature maps of the ECA module, respectively. $\sigma$ denotes the sigmoid function. $Conv^{1×1}$ denotes the convolution operation with a filter size of $1 × 1$. $GAP$ denotes for global average pooling operation.

As shown in Figs. 4A and 4B, the improved fused MBconv block (IFMBConv) and the improved MBConv block (IMBConv) used in this paper are shown schematically. The CFE module specific parameter information can be found in Table 1, which describes each phase of the DCT-FDE module in detail. The parameter Stride indicates whether the first convolution in the first IFMBConv block or IMBConv block in a phase consisting of IFMBConv or IMBConv compresses the feature map, when the step size is 1, the feature map size is unchanged, and when the step size is 2, the compressed feature map size is one half of the input feature map. The parameter Channel indicates the size of the feature map at the time of input to the current stage. The Layers parameter represents the number of times the IFMBConv block or IMBConv block is repeated. Note that the size of the feature map output at stage 7 is $7 × 7$ and the number of channels is 384.

# EXPERIMENTS

In this section, our method FFENet will conduct comparative experiments on three datasets, Clothing-8, fashion-mnist and Deepfashion, and verify the rationality of the structural design by conducting ablation experiments on the Clothing-8 dataset. Clothing 8 is our own small sample dataset built for complex scenes, while fashion-mnist and Deepfashion are public clothing datasets. The performance of our model in specific scenarios is verified on the Clothing-8 dataset, and the experiments on the fashion-mnist dataset and Deepfashion dataset are used to verify the performance of our model on regular large datasets.

## Implement details

System configuration information for the experimental platform. The system version is Windows 10, the processor is an Intel(R) Core(TM) i9-12900KF CPU @ 3.20 GHz and the GPU is an NVIDIA GeForce RTX 3090 Ti 24GB. Conda environment relies on python 3.8. The optimizer used is the SGD optimizer, which the initial learning rate is 0.01 and the decay coefficient of the optimizer is 0.0001. The input network has $224 \times 224$ image pixels, batch size is 8. GoogleNet, ResNet, DenseNet, EfficientNet, ConvNext, EfficientNetV2, ViT and SwinT were chosen to compare the classification of the models.

## Dataset
### Clothing-8.

The Clothing-8 dataset is an 8-category clothing dataset dedicated to the clothing category. We formed our initial dataset by collecting open source clothing images from Kaggle and shopping websites, combined with a small number of clothing images from the DeepFashion dataset. Through discussions on the collected images, we categorized them into eight general clothing categories. The Clothing 8 dataset comprises a training set with 4,550 examples and a validation set with 606 examples. The dataset includes the following eight categories: dress, jacket, pants, polo, shirt, tank top, t-shirt, and warmcloth, with corresponding counts of 613, 614, 731, 601, 571, 614, 722, and 691. When we trained, the ratio of the training set to the test set was 8:2.

### Fashion-MNIST.

Fashion-MNIST dataset is a clothing dataset with consistent backgrounds. The fashion-mnist dataset have 70,000 examples. Each example is a $28 \times 28$ gray-scale image associated with a label from 10 classes. The 10 categories are Angle Boot, Bag, Coat, Dress, Pullover, Sandal, Shirt, Sneaker, Trouser and T-shirt. The fashion-mnist dataset has the ratio of the training set to the test set is 6:1.

### DeepFashion.

DeepFashion dataset is a large-scale clothing dataset consisting of real-world scenarios, including buyer's show and seller's show. It contains 289,222 annotated fashion clothes images, of which 209,222 are used for training, 40,000 are used for verification, and the remaining 40,000 are test samples. Each image is labelled with 46 clothing categories. The ratio of training set, validation set, and test set in the dataset is approximately 5:1:1.

## Evaluation criterion

We usually call the prediction is correct and positive as true positive (TP). A false positive (FP) is a prediction that is false and positive. If the prediction os correct and the result is negative, it is called true negative (TN). A false negative (FN) is when the prediction is false and negative. Based on the above theories, the evaluation indexes of the text are as follows. Accuracy is the proportion of correct prediction results in total prediction, which specific calculation method is shown in Eq. (2). Precision is the percentage of positive predictions that are correct, which specific calculation method is shown in Eq. (3). Recall is the percentage of all positive events that correctly predicted the result, which specific calculation method is shown in Eq. (4). Model accuracy is equal to the number of correct predictions (NCP) of all kinds divided by the total number of verified pictures (TNVP), which specific calculation method is shown in Eq. (5). It is the parameter most often used to evaluate the quality of model training. The equations are as follows:

$$Accuracy = \frac{TP + TN}{TP + TN + FP + FN} \tag{2}$$

$$Precision = \frac{TP}{TP + FP} \tag{3}$$

$$Recall = \frac{TP}{TP + FN} \tag{4}$$

$$Model\ Accuracy = \frac{NCP}{TNVP} \tag{5}$$

where accuracy here is calculating the probability that a single category is correct, whereas model accuracy is calculating the probability that the entire model is correct. Accuracy, Precision, and Recall can only reflect the performance of individual categories. To provide a more intuitive assessment of model performance in terms of Accuracy, Precision, and Recall, we introduce mAccuracy, mPrecision, and mRecall to evaluate the effectiveness of each model. By averaging Accuracy, Precision, and Recall, we make the experimental results more comprehensible. Additionally, we incorporate the number of model parameters and model complexity as indicators to analyze our model in the ablation experiment.

## Comparison evaluation on Clothing8 dataset

The number of training rounds for our experiments on the Clothing-8 dataset is 100. The robustness of the model is very important, so we verify the performance of our model on the Clothing-8 validation set. The three indicators compared in Table 2 are Precision, Recall and Accuracy, and it can be seen that the effect of our model is better than these models. In this complex scene small sample dataset, it can be seen that the classification model based on convolution is better than the classification model based on tranformer. Firstly, in terms of the prediction mPrecision metric, our model outperforms the best EfficientNetV2 by 3.32%. Secondly, from the average recall, EfficientNetV2 has the best

**Table 1 The structure of the CFE module.**

| Stage | Operator | Input size | Stride | Channel | Layers |
|-------|----------|-----------|--------|---------|--------|
| 1 | Conv $3 \times 3$ | $224 \times 224$ | 2 | 48 | 4 |
| 2 | IFMBConv1 | $112 \times 112$ | 1 | 48 | 7 |
| 3 | IFMBConv2 | $112 \times 112$ | 2 | 48 | 7 |
| 4 | IFMBConv2 | $56 \times 56$ | 2 | 64 | 10 |
| 5 | IMBConv1 | $28 \times 28$ | 2 | 96 | 19 |
| 6 | IMBConv2 | $14 \times 14$ | 1 | 192 | 25 |
| 7 | IMBConv1 | $14 \times 14$ | 2 | 224 | 7 |

effect, but our model is 3.46% better than ConvNext. Finally, our model is also the best in terms of mAccuracy metric, our mAccuracy is 98.35%, which is 0.82% better than the best existing model EfficientNetV2 in the table.

Table 3 shows the comparison results of Model Accuracy between our model and other existing models on the Clothing-8 validate seen from Table 3 that ResNet, ConvNeXt, and EfficientNet V2 achieve preferably performance in the existing methods, and their accuracy rates are 90.01%, 89.46% and 87.79%, respectively. However, our model EfficientNet V2 achieves an accuracy of 93.4%, 3.39% better than the best model EfficientNetV2. The performance of our model on this small sample dataset of complex scenes is impressive. We verify our code on the DeepFashion dataset and replicate the effects of EfficientNet v2 and EfficientNet-B7 on this dataset, showing that our model FFENet still performs the best.

## Comparison evaluation on Fashion-MNIST dataset

The number of training rounds for our conduct experiments on the fashion-mnist dataset is 50. Our model is compared with some existing classification models for the same volume on the fashion-mnist test set, and these experiments are not pre-trained. Among them, ViT and SwinT adopt the largest size model, and ConvNeXt adopts base model, largest model used by EfficientNet and EfficientNetV2. Transformer works well on very large datasets, but the transformer model do not work well on fashion-mnist dataset, so we used the largest volume of Transformer classification model to compare with convolutional classification model.

As can be seen from Table 4, in terms of model accuracy, our model outperforms the best model EfficientNetV2-L by 0.69%. According to the mAccuracy metric, our model is 0.13% better than the best model ConvNext. Then, it also has good effects from the indicators of mRecall and mPrecision. Its mRecall and mPrecision are both 94.62%, 0.69% and 0.64% higher than the current best algorithm respectively. From these comparative experiments, we can see that our model works well even on datasets with simple backgrounds.

## Comparison evaluation on deepfashion dataset

In our comparison experiment on the DeepFashion dataset, we conducted 100 training iterations with a batch size of 16. Table 5 presents a comparison of the models used for clothing classification in the DeepFashion test set. Classic models like FashionNet are

**Table 2  Comparison of classification performance on the Clothing-8 validation set.** Results that surpass all competing methods are bold font. The upward arrow next to the parameter in the table indicates that the larger the parameter, the better.

| Model | Precision (%) ↑ | | | | | | | | mPrecision (%) ↑ |
|---|---|---|---|---|---|---|---|---|---|
| | Dress | Jacket | Pant | Polo | Shirt | T-shirt | Tank top | Warmcloth | |
| GoogleNet | 73.33 | 92.42 | 97.75 | 87.32 | 90.62 | **92.68** | 91.94 | 79.38 | 86.33 |
| Resnet-101 | 82.67 | 95.52 | 82.80 | 82.3 | 80.6 | 82.7 | 91.90 | 90.4 | 89.74 |
| DenseNet-201 | 78.57 | 95.71 | 98.75 | 78.21 | 84.72 | 82.93 | 81.82 | 82.02 | 85.34 |
| EfficientNet-B7 | 78.87 | 95.38 | 94.38 | 78.31 | 85.51 | 87.95 | 89.23 | 89.02 | 87.33 |
| ViT-L | 25.74 | 44.58 | 49.21 | 22.45 | 36.36 | 36.21 | 36.21 | 40.62 | 34.86 |
| Swin-L | **95.65** | 89.33 | 95.79 | 83.82 | 78.33 | 77.92 | 86.05 | 81.17 | 86.01 |
| ConvNext-L | 75.00 | **100.00** | 96.51 | 81.82 | 92.42 | 82.95 | 88.06 | 88.10 | 88.11 |
| EfficientNetV2-L | 78.26 | 96.97 | **100.00** | 82.28 | 94.03 | 86.36 | **95.16** | 88.64 | 90.21 |
| **FFENet**(ours) | 91.07 | **100.00** | 97.98 | **91.55** | **96.67** | 82.67 | 91.67 | **96.20** | **93.53** |
| Model | Recall (%) ↑ | | | | | | | | mRecall (%) ↑ |
| | Dress | Jacket | Pant | Polo | Shirt | T-shirt | Tank top | Warmcloth | |
| GoogleNet | 79.71 | 88.41 | 92.13 | 94.20 | 83.82 | 86.21 | 80.00 | 81.40 | 85.74 |
| Resnet-101 | 89.86 | 92.75 | 97.75 | 89.86 | 85.29 | 87.36 | 81.43 | 89.53 | 89.23 |
| DenseNet-201 | 79.71 | 97.10 | 88.76 | 88.41 | 81.71 | 78.16 | 77.14 | 84.88 | 85.48 |
| EfficientNet-B7 | 81.16 | 89.86 | 94.38 | 94.20 | 86.76 | 83.91 | 82.86 | 84.88 | 87.25 |
| ViT-L | 37.68 | 53.62 | 69.66 | 15.94 | 52.94 | 24.14 | 20.0 | 15.12 | 36.14 |
| Swin-L | 83.02 | 95.71 | 91.92 | 79.17 | 74.60 | 83.33 | 92.5 | 83.13 | 85.42 |
| ConvNext-L | 82.61 | 89.86 | 93.26 | 91.30 | 89.71 | 83.91 | 84.29 | 87.06 | 87.75 |
| EfficientNetV2-L | 78.26 | 92.75 | **97.75** | 94.20 | 92.65 | 87.36 | 84.29 | 91.76 | 89.88 |
| **FFENet**(ours) | **86.96** | **95.65** | **97.75** | **97.10** | **94.12** | **90.80** | **91.43** | **92.94** | **93.34** |
| Model | Accuracy (%) ↑ | | | | | | | | mAccuracy (%) ↑ |
| | Dress | Jacket | Pant | Polo | Shirt | T-shirt | Tank top | Warmcloth | |
| GoogleNet | 94.40 | 97.86 | 98.68 | 96.87 | 97.53 | 95.39 | 96.71 | 94.23 | 96.46 |
| Resnet-101 | 96.71 | 95.68 | 99.34 | 97.36 | 97.36 | **97.20** | 97.03 | 95.22 | 97.36 |
| DenseNet-201 | 95.22 | 99.18 | 98.19 | 95.88 | 97.03 | 94.56 | 95.39 | 95.22 | 96.33 |
| EfficientNet-B7 | 95.39 | 98.35 | 98.35 | 96.38 | 96.87 | 96.05 | 96.87 | 96.38 | 96.83 |
| ViT-L | 80.56 | 87.15 | 85.01 | 84.18 | 84.35 | 83.03 | 83.36 | 84.84 | 84.84 |
| Swin-L | 80.14 | 88.14 | 87.97 | 85.61 | 85.10 | 85.10 | 86.96 | 84.93 | 85.49 |
| ConvNext-L | 94.88 | 98.84 | 98.51 | 96.70 | 98.02 | 95.21 | 96.86 | 96.53 | 96.94 |
| EfficientNetV2-L | 95.05 | 98.84 | **99.67** | 97.03 | 98.51 | 96.20 | 97.69 | 97.19 | 97.52 |
| **FFENet**(ours) | **97.17** | **99.34** | **99.67** | **97.85** | **99.01** | 97.19 | **98.68** | **97.85** | **98.35** |

widely recognized in the field of fashion category classification. The citation of the top-k accuracy rate from the original DeepFashion paper in this study is based on the consistency of dataset division with the DeepFashion paper. Furthermore, we reproduce EfficientNet V2 and EfficientNet-B7 on the DeepFashion dataset, assessing their performance in the clothing classification task.

As shown in Table 5, our method FFENet consistently exhibits superior performance. In comparison to the specialized clothing classification model, FashionNet, our method achieves a 5.87% improvement in Top-3 accuracy and a 3.55% improvement in Top-5 accuracy. Moreover, our method outperforms the commonly used classfication model

**Table 3 Comparison of classification performance on the Clothing-8 validation set.** Results that surpass all competing methods are bold font. The upward arrow next to the parameter in the table indicates that the larger the parameter, the better.

| Model | Model accuracy (%) ↑ |
|---|---|
| GoogleNet | 85.83 |
| Resnet-101 | 89.46 |
| DenseNet-201 | 85.33 |
| EfficientNet-B7 | 87.31 |
| ViT-L | 36.24 |
| SwinT-L | 74.1 |
| ConvNext-L | 87.79 |
| EfficientNetV2-L | 90.01 |
| **FFENet** (ours) | **93.4** |

**Table 4 Comparison of classification performance on the Fashion-MNIST test set.** Results that surpass all competing methods are bold font. The upward arrow next to the parameter in the table indicates that the larger the parameter, the better.

| Model | Model accuracy (%) ↑ | mAccuracy (%) ↑ | mRecall (%) ↑ | mPrecision (%) ↑ |
|---|---|---|---|---|
| GoogleNet | 88.18 | 97.63 | 88.18 | 88.34 |
| ResNet | 90.00 | 98.00 | 90.00 | 90.05 |
| DenseNet | 91.11 | 98.22 | 91.11 | 91.15 |
| EfficientNet | 93.87 | 98.78 | 93.89 | 93.88 |
| ViT | 86.70 | 97.33 | 86.66 | 86.64 |
| SwinT | 90.08 | 98.07 | 90.35 | 90.33 |
| ConvNext | 93.86 | 98.75 | 93.76 | 93.73 |
| EfficientNetV2 | 93.93 | 98.79 | 93.93 | 93.98 |
| **FFENet** (ours) | **94.62** | **98.92** | **94.62** | **94.62** |

**Table 5 Comparison of classification performance on the DeepFashion test set.** Results that surpass all competing methods are shown in bold.

| Model | Category | |
|---|---|---|
| | Top-3 (%) | Top-5 (%) |
| WTBI (*Liu et al., 2016*) | 43.73 | 66.26 |
| DARN (*Liu et al., 2016*) | 59.48 | 79.58 |
| FashionNet+100 (*Liu et al., 2016*) | 47.38 | 70.57 |
| FashionNet+500 (*Liu et al., 2016*) | 57.44 | 77.39 |
| FashionNet+Joins (*Liu et al., 2016*) | 72.30 | 81.52 |
| FashionNet+Poselets (*Liu et al., 2016*) | 75.34 | 84.87 |
| FashionNet (*Liu et al., 2016*) | 82.58 | 90.17 |
| EfficentNet-B7 | 87.87 | 93.47 |
| EfficientNetV2-L | 87.89 | 93.42 |
| **FFENet** (ours) | **88.45** | **93.72** |

**Table 6 Comparison of classification performance on the Clothing-8 dataset.** ACs stands for channel information for adjusting the constructed network. Results that surpass all competing methods are bold font. The upward arrow next to the parameter in the table indicates that the larger the parameter, the better. The downward arrow next to the parameter in the table indicates that the smaller the parameter, the better.

| DCT-FDE module | ACs | ECA | Model accuracy (%) ↑ | Params (M) ↓ |
|---|---|---|---|---|
| | | | 90.01 | 117.24 |
| ✓ | | | 90.76 | 117.26 |
| ✓ | ✓ | | 91.91 | 117.36 |
| ✓ | ✓ | ✓ | **93.40** | **95.16** |

EfficientNetV2-L by 0.56% in terms of Top-3 accuracy and by 0.3% in terms of Top-5 accuracy.

## Ablation study

Our ablation experiments are conducted on the Clothing-8 dataset, and the number of training rounds is set to 100. The other experimental settings are the same as those in subsection 'Implement Details'. We choose MBConv block and fusion MBConv block to build the network structure, and the specific information can be found in subsection 'Clothing Feature Extraction Module'. Table 6 shows our improvement process. We add DCT-FDE module and can see a 0.75% improvement in accuracy. We think about why the accuracy is not improved a lot. Considering that the number of channels of our DCT-FDE module output feature map is 48, we adjust the number of input and output channels of the first and second stages of the network to be 48, so that the information of the feature map output by our DCT-FDE module can be fully learned. The number of channels in the previous first and second stage are less than 48, so the learned feature map information must be insufficient. By adjusting the number of channels in the network we get another 1.15% improvement. Because the SE module inside the MBConv block and fused MBConv block has the operation of compression channel, so we replace the SE block with the ECA module. ECA module is also a channel attention mechanism, but the ECA module has no operation to compress the channel. Experiments show that the model accuracy is improved by 1.49% after replacing the SE module with the ECA module. The final model parameters decreased by 22.08M compared with the initial model.

Table 7 shows our other ablation experiment, in which the influence of DCT block size on the accuracy of the final model is discussed. According to the data in the table, we can find that the accuracy rate of $2 \times 2$ block is lower than that of $4 \times 4$ block. Meanwhile, compared with $2 \times 2$ block, the number of parameters and computational complexity in $4 \times 4$ block are not much improved. From the data in the table, if the $8 \times 8$ DCT block size is used, the accuracy is also decreased a little compared with the $4 \times 4$ block size, and the number of parameters and the computational complexity are increased, so we choose $4 \times 4$ block as our block size.

**Table 7 Comparison of classification performance on the Clothing-8 dataset.** Results that surpass all competing methods are bold font. The upward arrow next to the parameter in the table indicates that the larger the parameter, the better. The downward arrow next to the parameter in the table indicates that the smaller the parameter, the better.

| DCT block size | Model accuracy (%) ↑ | Params (M) ↓ | GFLOPs ↓ |
|---|---|---|---|
| 2 ×2 | 90.586 | **117.25** | **12.33** |
| 4 ×4 | **90.760** | 117.26 | 12.46 |
| 8 ×8 | 90.759 | 117.30 | 12.98 |

## CONCLUSIONS

In this work, our primary objective was to enhance the performance of clothing classification in complex scenes, recognizing that clothing classification heavily relies on texture and contour information. Previous studies, along with our own experiments, have confirmed that different frequency bands in the frequency domain store distinct image information, including texture, contour, and other relevant details. By extracting and leveraging this information through a feature extraction module based on discrete cosine transform (DCT), combined with a fully convolutional backbone algorithm, we propose a clothing classification network known as the frequency-spatial feature enhancement network. Our algorithm effectively improves the accuracy of clothing classification. Extensive experiments were conducted on three datasets: Clothing-8, Fashion-MNIST, and DeepFashion. The results demonstrate the superior performance of our constructed network in both clothing dataset with complex scenes and clothing dataset with consistent scenes.

While our method exhibits promising results, it is important to acknowledge its limitations. For instance, our approach may not provide a significant boost for clothing images with simple backgrounds. Addressing these limitations will be a focus of our future work, aiming to further refine and enhance the applicability of our proposed method.

### Funding

This work was supported by the National Natural Science Foundation of China (No. 62202346), the Hubei key research and development program (No.2021BAA042), the open project of engineering research center of Hubei province for clothing information (No. 2022HBCI01), the Wuhan applied basic frontier research project (No. 2022013988065212), MIIT's AI Industry Innovation Task unveils flagship projects (Key technologies, equipment, and systems for flexible customized and intelligent manufacturing in the clothing industry), and the Hubei science and technology project of safe production special fund (Scene control platform based on proprioception information computing of artificial intelligence). The funders had no role in study design, data collection and analysis, decision to publish, or preparation of the manuscript.

### Grant Disclosures

The following grant information was disclosed by the authors:

National Natural Science Foundation of China: 62202346.
Hubei key research and development program: 2021BAA042.
Open project of engineering research center of Hubei province for clothing information: 2022HBCI01.
Wuhan applied basic frontier research project: 2022013988065212.
MIIT's AI Industry Innovation Task unveils flagship projects (Key technologies, equipment, and systems for flexible customized and intelligent manufacturing in the clothing industry).
Hubei science and technology project of safe production special fund (Scene control platform based on proprioception information computing of artificial intelligence).

## Competing Interests

The authors declare that there are no competing interests.

## Author Contributions

- Feng Yu conceived and designed the experiments, performed the experiments, analyzed the data, performed the computation work, prepared figures and/or tables, authored or reviewed drafts of the article, he provides fund support for the project, and approved the final draft.
- Huiyin Li conceived and designed the experiments, performed the experiments, analyzed the data, performed the computation work, prepared figures and/or tables, authored or reviewed drafts of the article, and approved the final draft.
- Yankang Shi performed the experiments, analyzed the data, performed the computation work, prepared figures and/or tables, and approved the final draft.
- Guangyu Tang performed the experiments, analyzed the data, performed the computation work, prepared figures and/or tables, and approved the final draft.
- Zhaoxiang Chen conceived and designed the experiments, authored or reviewed drafts of the article, and approved the final draft.
- Minghua Jiang conceived and designed the experiments, authored or reviewed drafts of the article, he provides fund support for the project, and approved the final draft.

## Data availability

The code is available on GitHub and Zenodo:

- https://github.com/jasminelhy/clothing-classfication/tree/main/FFENet

- jasminelhy. (2023). jasminelhy/clothing-classfication: First release of my code (V1.0.0). Zenodo. https://doi.org/10.5281/zenodo.8119040

The data is available at Figshare:

Li, Huiyin (2023). Clothing8. figshare. Dataset. https://doi.org/10.6084/m9.figshare.22220011.v1.

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
