# Peer review of "FFENet: frequency-spatial feature enhancement network for clothing classification"

_PeerJ Computer Science, doi:10.7717/peerj-cs.1555_

## Round 0.1 · original submission · Major Revisions

Please address the reviewer comments and concerns.

Reviewer 1 ·

Basic reporting

In this manuscript, the authors improve upon existing convolutional networks for clothing classification. In particular, they develop a discrete-cosine transform based approach to extract information about spatial frequencies in images of clothing and feed this information as additional features to convolutional network.

The idea is interesting and the approach more or less valid, but the English used in this paper needs improvement. I suggest getting the paper proof read by a more proficient English speaker before resubmitting with revisions. Some examples of text pieces that need to be fixed/changed/improved:

1. The first line of the abstract is: "Clothing analysis has been widely concerned by people, and clothing classification, as one of the most basic technologies, plays a very important role in the field of clothing analysis" -this is structurally incorrect

2. Line 18: "The learning of clothing features in complex scenes is disturbed" - I do not understand what "disturbed" means here

3. When describing related work in the section under "Related Work", it would be more appropriate to write in the past tense, than in the present tense. For example, instead of saying "First, for DCT, Qin et al (2021) studies the effect of partially compressed input images....", replace "studies" with "studied". The entire section should be use the past tense instead of present tense.

I have highlighted more such examples in the annotated manuscript. It's possible I missed some, which is why I suggest getting the manuscript proof read by a native English speaker or a more proficient English speaker

Experimental design

The experiment design is reasonable but there are some issues:

1. The authors use a custom built corpus called, Clothing8, in addition to the publicly available Fashion-MNIST. How was this corpus created? What was the distribution of the 8 categories, i.e., how many items in each (Lines 240-244)? The distribution of items is very important, since if the category counts are imbalanced, the measures of accuracy can be non-reliable.

2. In Table 4, the authors report something called "mRecall", "mPrecision" or "mAccuracy". What is the significance of that over the common Recall/Accuracy/Precision?

Validity of the findings

In general, the gains reported here seem valid but the information about class distributions, as I mentioned in the section on "Experimental Design", are very important

Annotated reviews are not available for download in order to protect the identity of reviewers who chose to remain anonymous.
Cite this review as

Reviewer 2 ·

Basic reporting

This paper proposes a clothing classification network based on frequency-spatial domain conversion, which combines frequency-domain information with spatial information without compressing feature map channels. It extracts high and low-frequency information from the feature maps and transforms this information from the frequency domain into a spatial domain image. Experiments are conducted on clothing 8 and fashion-mnist datasets to show the effectiveness of the proposed method.

The topic of the paper is interesting and fits the scope of the Journal. The new problem setting is interesting and it is worth investigating this new setting. However, the relation to prior work is not discussed sufficiently well. The paper writing could be significantly improved. There are too many parts of the paper that need to be rewritten to better guide the reader.

Experimental design

The ablation study is comprehensive and demonstrates the importance of each novel component in the proposed method.
However, the experiments are less convincing due to some reasons:
(1) As claimed in the paper, the proposed method attempts to handle the problem of poor clothing classification because of the complexity and variety of clothing scenes. However, the datasets used for experiments seem to have simple scenes and backgrounds.
(2) The proposed method is only compared to generic classification models, not models for clothing-specific classification task, such as
(a) Y. Zhang, P. Zhang, C. Yuan and Z. Wang, "Texture and Shape Biased Two-Stream Networks for Clothing Classification and Attribute Recognition," 2020 IEEE/CVF Conference on Computer Vision and Pattern Recognition (CVPR), Seattle, WA, USA, 2020, pp. 13535-13544.
(b) S. C. Hidayati, C. -W. You, W. -H. Cheng and K. -L. Hua, "Learning and Recognition of Clothing Genres From Full-Body Images," in IEEE Transactions on Cybernetics, vol. 48, no. 5, pp. 1647-1659, May 2018.
(c) Z. Liu, P. Luo, S. Qiu, X. Wang and X. Tang, "DeepFashion: Powering Robust Clothes Recognition and Retrieval with Rich Annotations," 2016 IEEE Conference on Computer Vision and Pattern Recognition (CVPR), Las Vegas, NV, USA, 2016, pp. 1096-1104.

Validity of the findings

Comparisons with the related work are missing. The experiments section doesn't sufficiently describe the claim. Additional experiments (see above) is in need.

Cite this review as

---

## Round 0.2 · accepted · Accept

Thank you for addressing the reviewer comments.

Reviewer 1 ·

Basic reporting

The authors have addressed the changes I requested/suggested. One minor comment is to add "respectively" at the end of reporting the counts of each type of clothing in line 309

Experimental design

NA

Validity of the findings

NA

Cite this review as